# Learning Causal Biological Networks with Parallel Ant Colony Optimization Algorithm

**DOI:** 10.3390/bioengineering10080909

**Published:** 2023-07-31

**Authors:** Jihao Zhai, Junzhong Ji, Jinduo Liu

**Affiliations:** Beijing Municipal Key Laboratory of Multimedia and Intelligent Software Technology, Beijing Institute of Artificial Intelligence, Faculty of Information Technology, Beijing University of Technology, Beijing 100124, Chinajjz01@bjut.edu.cn (J.J.)

**Keywords:** causal biological networks, causal brain networks, causal protein signaling networks, parallel ant colony optimization, pheromone fusion, CBNs fusion

## Abstract

A wealth of causal relationships exists in biological systems, both causal brain networks and causal protein signaling networks are very classical causal biological networks (CBNs). Learning CBNs from biological signal data reliably is a critical problem today. However, most of the existing methods are not excellent enough in terms of accuracy and time performance, and tend to fall into local optima because they do not take full advantage of global information. In this paper, we propose a parallel ant colony optimization algorithm to learn causal biological networks from biological signal data, called PACO. Specifically, PACO first maps the construction of CBNs to ants, then searches for CBNs in parallel by simulating multiple groups of ants foraging, and finally obtains the optimal CBN through pheromone fusion and CBNs fusion between different ant colonies. Extensive experimental results on simulation data sets as well as two real-world data sets, the fMRI signal data set and the Single-cell data set, show that PACO can accurately and efficiently learn CBNs from biological signal data.

## 1. Introduction

The rapid development of science and technology yields a huge amount of biological signal data and also drives the development of related fields in biological systems [1,2]. For example, the invention of functional magnetic resonance imaging (fMRI) facilitates to learn causal brain networks from brain activity [3,4,5], the creation of intracellular multicolor flow cytometry allows more quantitative simultaneous observations of multiple signaling molecules in many thousands of individual cells and making it easier to infer causal protein signaling networks among protein biomolecules [6,7] and the invention of Single-cell RNA sequencing (ScRNA-seq) yields large amounts of gene expression data, bringing new research perspectives to learn the causal regulatory relationships between different genes [8,9]. A causal biological network (CBN) is a set of nodes and directed edges that can succinctly represent the causal relationships between different types of biological nodes mentioned above [10]. Learning CBNs accurately and efficiently from biological signaling data has been an important issue in recent years [11], and is important for a deeper understanding of the underlying principles in biological mechanisms. In recent years, many CBN learning methods have been proposed, which can be broadly classified into two categories, one based on traditional machine learning and the other on deep learning methods with complex model structures.

Traditional machine learning methods include Linear non-Gaussian Acyclic Model (LiNGAM) [12] based methods, Bayesian Network (BN) based methods [13] and Granger Causality (GC) [14] based methods. etc. Recently, Wei et al. [15] proposed a method for learning CBNs based on BN with pruning strategies. Zhang et al. [16] proposed a CBNs learning method based on truncated matrix power iteration. Gao et al. [17] proposed a Gaussian model for optimal learning of CBNs and showed significant performance improvement. The advantages of these methods are simple models, relative flexibility, and short running times, but the learned CBNs are often not accurate enough and easily fall into local optima.

With the rapid development of deep learning, many deep learning methods are also successfully used to learn CBNs. Yu et al. [18] proposed a graph neural network-based CBNs learning method and successfully applied it to learn causal protein signaling networks. Fan et al. [8] used 3D convolutional neural networks in successfully learning more accurate causal gene regulatory networks. Lu et al. [19] proposed a deep reinforcement learning-based framework and Liu et al. used generative adversarial network [20] and recurrent generative adversarial network [21] for learning causal brain networks. Compared with traditional machine learning methods, the accuracy of learning CBNs from biological signal data using deep learning methods will be improved, but it will cost a lot of time because the model structure is mostly complex.

To solve the above-mentioned problems of existing methods, in this paper, we propose a novel CBNs learning algorithm called parallel ant colony optimization (PACO), which utilizes a parallel ant colony optimization algorithm to learn CBNs from biological signal data. The PACO algorithm consists of three main phases: initialization, parallel ant colony optimization, and pheromone fusion and CBNs fusion phase. During the initialization phase, PACO initializes the parallel ant colony and sets some initial parameters for the ant colonies. In the parallel ant colony optimization phase, the K2 metric is used to measure the quality of the learned CBNs and guides the ant colony search. PACO employs multiple ant colonies to learn the best CBN with the highest K2 metric in parallel. In the pheromone fusion and CBNs fusion phase, all ant colonies are guided to perform a more accurate search by sharing pheromones from the colony with the highest K2 metric to other colonies. Finally, PACO obtains the best CBN from all ant colonies according to the extraction rule. Extensive experimental results on simulation data sets as well as on two real-world data sets, the fMRI signal data set and the Single-cell data set, show that PACO outperforms other state-of-the-art or classical methods in learning CBNs from biological signal data. The main contributions of this paper can be summarized as follows:To the best of our knowledge, this is the first study to employ a parallel ant colony optimization algorithm to learn CBNs from biological signal data. The incorporation of parallelization allows for more accurate and efficient learning of CBNs, which will provide a significant reference for the causal discovery and bioinformatics fields.PACO incorporates the parallel ant colony optimization and information fusion strategy. This approach not only enhances the algorithm’s efficiency and reduces time complexity, but also facilitates the extraction of shared information from multiple data sets, thereby improving the accuracy of learn CBNs and more fully utilizes global information, effectively reducing the probability of falling into a local optima.Numerous experiments conducted on simulation data sets, fMRI signal data sets and Single-cell data set have demonstrated that the proposed method is capable of learning CBNs from different biological signal data, thereby improving inference performance, which has significant implications for a deeper understanding of the underlying causal relationships in biological systems.

## 2. Related Work

### 2.1. Causal Biological Networks

The CBNs learned from different types of biological signal data can be specifically subdivided into many types, such as causal brain networks, causal protein signaling networks, causal gene regulatory networks and other CBNs, and we will describe the related work of causal brain networks and causal protein signaling networks in detail in the following. Table 1 shows the introduction of different CBN learning methods.

#### 2.1.1. Causal Brain Networks

Causal brain networks consist of multiple brain nodes and causal interactions between different nodes, and accurate learning of causal brain networks is valuable for understanding the functioning of brain cognition and gaining insight into the pathogenesis of brain diseases [32,33]. In recent years, many studies have emerged to learn causal brain networks from fMRI signal data, Friston et al. [22] first proposed a spectral dynamic causal modeling for learning causal brain networks from fMRI signal data. Zhang et al. [30] first proposed a amortization transformer model for learning causal brain networks from fMRI signal data. Li et al. [28] and Razi et al. [34] extended the model to learn the causal brain networks on large-scale brain regions from fMRI signal data. Ji et al. first proposed to learn causal brain networks using an artificial immune algorithm (AIA) [25] and a recurrent generative adversarial network (RGAN) model [21], with greatly performance. Li et al. [35] explored the dynamic abnormalities of brain function in Parkinson’s disease and the pathophysiological significance behind them by constructing causal brain networks from fMRI signal data.

#### 2.1.2. Causal Protein Signaling Networks

Causal protein signaling networks consist of multiple protein biomolecule nodes and causal relationships between different nodes. Learning causal protein signaling networks accurately from Single-cell data is important for understanding the causal relationships of biomolecules in cells and for gaining insight into the pathogenesis of cell-based diseases. Recently, Zhu et al. [26] designed a causal discovery model based on reinforcement learning that employs a reinforcement learning framework for learning causal protein signaling networks. Zheng et al. [23] first transformed the causal discovery problem from a combinatorial optimization problem to a continuous optimization problem by proposing a continuous optimization structure approach (NoTears) and successfully used it for learning causal protein signaling networks. Baek et al. [27] proposed to learn causal protein signaling networks using a three-track neural network. Li et al. [31] first proposed to learn causal protein signaling networks based on the Deconfounded Functional Structure Estimation. Squires et al. [29] proposed using latent factor causal models to learn causal protein signaling networks. Whitaker et al. [36] discussed the effect of p38 MAPK protein biomolecules on the relationship between cell cycle and apoptotic signaling pathways by constructing and analyzing the BCL2 family of causal protein signaling networks.

### 2.2. Ant Colony Optimization Algorithm

The ant colony optimization algorithm was originally proposed by Dorigo et al. [37] based on the intelligent behavior of ant colonies during the foraging process. After more than a decade of development, the ant colony algorithm has become one of the most effective algorithms for solving combinatorial optimization problems in swarm intelligence and has been widely used in various fields. Liu et al. [24] first used the ant colony algorithm to learn the effective connectivity of causal brain networks from fMRI signal data and to quantitatively characterize the strength of the connectivity. Liang et al. [38] proposed an improved context-based ant colony optimization algorithm and applied it to travel route planning. However, the tendency to fall into local optima is still one of the main factors limiting the performance of the algorithm.

## 3. The Parallel Ant Colony Optimization Algorithm

In this section, we will introduce a new algorithm to learn CBNs more accurately and efficiently from biological signal data.

### 3.1. Main Idea

To accurately and efficiently learn CBNs from biological signal data, we propose a novel algorithm called the parallel ant colony optimization (PACO). The PACO algorithm comprises three phases: initialization, parallel ant colony optimization, and pheromone fusion and CBNs fusion. Specifically, the PACO algorithm is a score-and-search approach for learning CBNs from biological signal data, utilizing the K2 metric to evaluate the quality of CBNs and guide the parallel ant colony to search for the global optimal CBNs. Additionally, we introduce a new information fusion mechanism that merges and updates the pheromones of all colonies after the completion of the same iteration of all ant colonies, serving as the initial pheromones of the colonies in the next iteration. When all iterations are completed, the optimal CBNs learned by all colonies are merged into an adjacency matrix. The final CBN is obtained by setting the extraction rules. Figure 1 illustrates the flowchart of the PACO algorithm.

### 3.2. Initialization

In the parallel ant colony optimization algorithm, a CBN can be represented as G=<V,E>, where *V* is a set of biological nodes and *E* is a set of arcs with each arc representing a causal interaction between two biological nodes. A CBN uses a graph structure and a set of parameters to encode uniquely the joint probability distribution of the domain variables $X=\{X_{1},X_{2},X_{3},\cdot \cdot \cdot X_{n}\}:$(1)$$P\left(X_{1},X_{2},X_{3},\cdot \cdot \cdot X_{n}\right)=\prod_{i=1}^{n}P\left(X_{i}|\prod \left(X_{i}\right)\right).$$

First we initialize *N* ant colonies with Num ants in each colony, then we initialize an empty CBN $G_{i}\left(0\right)\left(1\leq i\leq N\right)$ for each colony. Since ants produce pheromones and the concentration of pheromones changes continuously during the movement, we need to initialize a pheromone matrix for each colony. Finally, we need to initialize some parameters for the algorithm, such as the number of iterations NC, the number of iterations $l_{step}$ for local search, the initial information concentration $\tau_{0}$, etc. Additionally, we employ the K2 metric in PACO to evaluate the quality of CBNs. The K2 metric is a famous evaluation measure for learning CBNs from biological signal data, and the initial expression for the K2 metric is:(2)$$P\left(G,D\right)=P\left(G\right)\cdot \prod_{i=1}^{n}\prod_{j=1}^{q_{i}}\frac{(r_{i}-1)!}{(N_{ij}+r_{i}-1)!}\prod_{k=1}^{r_{i}}N_{ijk}!$$
where *D* is a given training set, *G* is a possible CBN, $r_{i}$ is the number of possible values of the variable $X_{i}$, $q_{i}$ is the number of possible configurations for the variables in $\prod (X_{i})$, and $N_{ijk}$ is the number of cases in *D* where $X_{i}$ has its *k*th value and $\prod (X_{i})$ is instantiated to its *j*th value.

### 3.3. Parallel Ant Colony Optimization

In this phase of the search process, we open *N* threads corresponding to *N* ant colonies searching *N* biological signal data sets in parallel. In PACO, each ant k(k=1,2,···Num) in *N* ant colonies starts from the empty CBN $G_{i}\left(0\right)\left(1\leq i\leq N\right)$ and increases one arc at a time until it is impossible to make the K2 metric of the CBNs higher by adding one arc. At time *t*, the probabilistic transition rule that an ant selects a directed arc $a_{ij}$ between two biological nodes $X_{i}$ and $X_{j}$ from the current set of candidate arcs is defined as:(3)$$a_{i,j}=\left\{\begin{array}{c} arg\;max_{i,j\in DA_{k}\left(t\right)}\left\{\left[\tau_{ij}\left(t\right)\right]\cdot {\left[\eta_{ij}\left(t\right)\right]}^{\beta }\right\},if\;q\leq q_{0},\\ a_{I,J},\;\;\;\;\;\;\;\;\;otherwise, \end{array}\right.$$
where $\tau_{ij}\left(t\right)$ is the pheromone concentration, $\eta_{ij}\left(t\right)$ represents the heuristic information of $a_{i,j}$, and β is the weighted coefficient which controls $\eta_{ij}\left(t\right)$ to influence the selection of arcs. $DA_{k}\left(t\right)\left(i,j\in DA_{k}\left(t\right)\right)$ is the set of all candidate arcs whose heuristic information is larger than zero; $q_{0}$ (0 $\leq q_{0}<$ 1) is an initial parameter that determines the relative importance of exploitation versus exploration (exploitation means selecting arcs by pheromone intensity and heuristic information, and exploration means global random selecting arcs); *q* is a random number uniformly sampled in [0,1]; and *I* and *J* are a pair of biological nodes randomly selected according to the probability in the following way:(4)$$p_{i,j}^{k}\left(t\right)=\left\{\begin{array}{c} \frac{{\left[\tau_{ij}\left(t\right)\right]}^{\alpha }\cdot {\left[\eta_{ij}\left(t\right)\right]}^{\beta }}{\sum_{r,l\in DA_{k}\left(t\right)}{\left[\tau_{rl}\left(t\right)\right]}^{\alpha }]\cdot {\left[\eta_{rl}\left(t\right)\right]}^{\beta }},\;if\;i,j\in DA_{k}\left(t\right),\\ 0,\;\;\;\;\;\;\;\;\;otherwise, \end{array}\right.$$
where α denotes the relative importance of $\tau_{rl}\left(t\right)$ left by ants. The heuristic function $\eta_{ij}$ is defined as follows:(5)$$\eta_{ij}\left(t\right)=\omega \cdot f(X_{i},\prod \left(X_{i}\cup X_{j}\right)-f\left(X_{i},\prod \left(X_{i}\right)\right)$$
where ω is a weighted factor concerned with the arc connecting intensity whose value is defined as:(6)$$\omega =1+Inf(X_{i},X_{j})$$
where $Inf(X_{i},X_{j})$ represents the mutual information between the two biological nodes $X_{i}$ and $X_{j}$. Because the mutual information $Inf(X_{i},X_{j})$ can objectively reflect whether the two biological nodes in a CBN are dependent and how much the dependency is, thus when the dependency intensity is stronger and the score-increase is larger, the heuristic information becomes greater, and vice versa. The mutual information between two biological nodes $X_{i}$ and $X_{j}$ is defined as:(7)$$Inf\left(X_{i},X_{j}\right)=\sum_{x_{i},x_{j}}P\left(x_{i},x_{j}\right)log\frac{P(x_{i},x_{j})}{P\left(x_{i}\right)P\left(x_{j}\right)}$$
after each iteration of the ant colony is performed, the PACO algorithm will carry out the pheromone updating process, which includes local and global updating steps. For the local optimization process, when an ant selects an arc $a_{ij}$, the pheromone level of the corresponding arc is changed in the following way:(8)$$\tau_{ij}\left(t+1\right)=\left(1-\rho \right)\tau_{ij}\left(t\right)+\rho \tau_{0}$$
where 0<ρ≤1 is a parameter that controls the pheromone evaporation.

### 3.4. Pheromone Fusion and CBNs Fusion

The above search process is performed by *N* colonies in parallel. Each colony finds the best solution from all feasible CBNs learned so far by means of the K2 metric, and performs the global updating for each arc of the current best CBN. The global updating rules at the $t_{th}$ iteration are shown in Equations (9) and (10):(9)$$\tau_{ij}\left(t\right)=\left(1-\rho \right)\tau_{ij}\left(t-1\right)+\rho \Delta \tau_{ij}\left(t\right)$$
(10)$$\Delta \tau_{ij}\left(t\right)=\left\{\begin{array}{c} \frac{1}{\left|f(G^{+}:D)\right|},\;if\;a_{ij}\in G^{+}\\ \tau_{ij}\left(t-1\right),\;otherwise \end{array}\right.$$

When all *N* colonies completed the above process, the algorithm enters the pheromone fusion and CBNs fusion phase, which is divided into two parts: when the number of iterations is not satisfied, the algorithm selects the pheromone matrix $M_{max}$ corresponding to the CBN with the highest K2 metric from all ant colonies, and updates the pheromone matrices $M_{1}$ to $M_{N}$ of all ant colonies to $M_{max}$ to guide the continued search of the ant colony during the next iteration. When the number of iterations reaches NC, the algorithm merges all the optimal CBNs learned by *N* ant colonies into the adjacency matrix *G*. Then we set the new extraction rule such that $G_{ij}=1,\left(1\leq i,j\leq N\right)$ when the value $G_{ij}\geq 50\%\cdot N$, otherwise $G_{ij}$ = 0. Finally, we extract the optimal CBN $G^{{}^{\prime }}$ learned from the *N* biological signal data sets.

Based on the above description, the termination process of PACO is as follows: when the current iteration number *t* reaches the preset iteration number NC, PACO merges the *N* CBNs learned by the parallel ant colony, and then extracts and outputs the optimal CBN $G^{{}^{\prime }}$ according to the designed rules, and the algorithm terminates.

### 3.5. Algorithm Description and Analysis

The PACO in this paper consists of three main phases: initialization, parallel ant colony optimization, and pheromone fusion and CBNs fusion phase, which are summarized in Algorithm 1. For the initialization phase, first, the PACO algorithm initialize *N* ant colonies and opens *N* threads for *N* colonies simultaneously. Then PACO generates an initial set of empty CBNs $G_{i}\left(0\right)$(1≤i≤N) and set some parameters for each colony. Finally, *N* biological signal data sets are input to the algorithm. For the parallel ant colony optimization phase, all ants in *N* colonies perform the search CBNs in parallel, starting with one empty CBN per ant and adding one arc at a time until the CBN cannot be constructed to have a higher K2 metric. The CBN learned is then locally optimized using Optimazation(), a function that uses the standard addition, deletion, and inversion operators for arcs. At this point, each ant colony obtains the optimal CBN for the current number of iterations *t* and updates the global pheromone. For the pheromone fusion and CBNs fusion phase, the algorithm selects the pheromone matrix $M_{max}$ corresponding to the CBN with the highest K2 metric from the *N* CBNs learned in the parallel ant colony optimization phase, and then updates $M_{max}$ to the pheromone matrix $M_{1}$ to $M_{N}$ to guide each ant colony to continue the search in the next iteration. When the NC iterations are completed, the algorithm merges *N* optimal CBNs learned by *N* ant colonies and extracts the optimal CBN $G{}^{\prime }$ according to the extraction rules we designed.

**Algorithm 1:** PACO

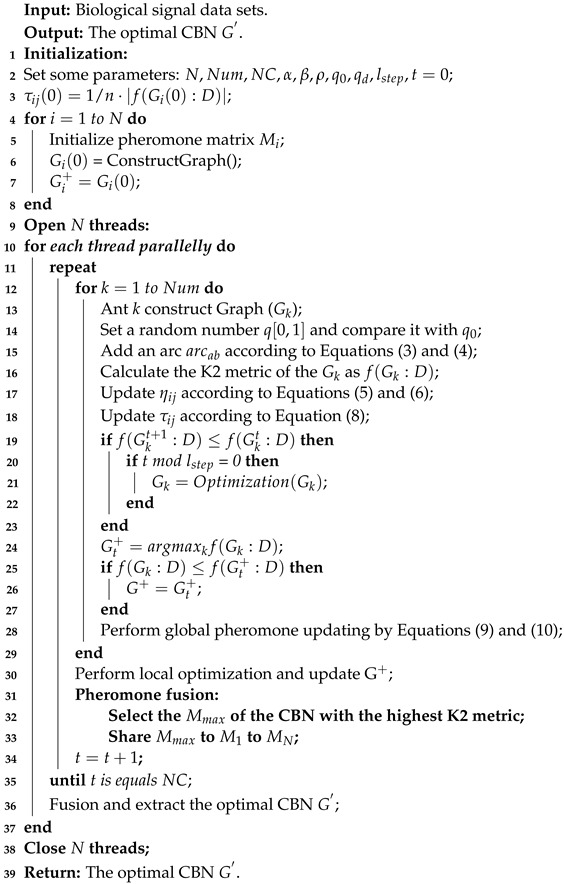



Based on the description of Algorithm 1, the complexity of PACO can be simply analyzed as follows: let the algorithm input *N* data sets, NC is the number of iterations, and the number of ants per colony is Num. Then the time complexity of PACO can be expressed as $O\left(N\right)+O\left(NC\right)\cdot O\left(3\cdot Num+N\right)\approx O\left(n^{2}\right)$. The time complexity of the non-parallel Ant Optimization Colony (ACO) algorithm can be expressed as $O\left(1\right)+O\left(N\right)\cdot O\left(NC\right)\cdot O\left(3\cdot Num\right)\approx O\left(n^{3}\right)$. It is obvious that PACO increases the time cost O(N) mainly in the pheromone fusion and CBNs fusion process compared to ACO, but for learning CBNs with multiple biological signal data sets, the time cost of PACO is much less than that of ACO, and the advantage becomes more and more obvious the larger the number *N* of biological signal data sets is.

## 4. Experimental Result of Learning CBNs

### 4.1. Data Description

#### 4.1.1. Simulation Data Sets

The generation of the simulation data sets is briefly described here, and more details can be found in [26]. Given a number of nodes *v*, we generate a d×d upper triangular matrix as the graph binary adjacency matrix, in which the upper entries are sampled independently from Bern(0.5). We assign edge weights independently from Unif ([−1.5, −0.5] ∪ [0.5, 1.5]) to obtain a weight matrix $W\in R^{d\times d}$, and then sample $x=W^{T}x+\lambda \in R^{d}$ from both Gaussian and non-Gaussian noise models. We choose unit noise variance in both models and use *m* = 200 samples as the sub-data set. The variables are then randomly ordered. Finally, we generate 10 simulation data sets sim1 to sim10 and each simulation data set consists of different numbers of nodes (*v* = 5, 10, 30, 50, 100) and different numbers of sub-data sets (*N* = 20, 50) in series. The input of each colony in PACO corresponds to each sub-data set of a simulation data set. Other algorithms take a complete simulation data set as input.

#### 4.1.2. fMRI Signal Data Sets

A causal brain network for representing the effective connectivity of different brain regions can be represented by a CBN. The Smith data set [39] is a set of fMRI signal data sets given by Smith et al. The data set is availableat http://www.fmrib.ox.ac.uk/datasets/netsim/index.html (accessed on 11 June 2023), which can be used to verify the accuracy of different methods to identify functional and effective brain connectivity and contains 28 data sets. Each data set contains different number of brain regions (Nodes), scan time (Session), repetition time of the pulse, noise, and several other influencing factors. We select the second of 28 data sets, which has 10 nodes and contains 50 subjects with 11 edges in the ground-truth to validate the performance of our method.

#### 4.1.3. Single-Cell Data Sets

Learning causal protein signaling networks from human immune Single-cell data can also be considered as the process of learning CBNs. The real multi-parameter fluorescence-activated cell sortera data set [6] to learn causal protein signaling networks based on expression levels of proteins and phospholipids, and the Single-cell data sets are availableat https://www.science.org/doi/10.1126/science.1105809#supplementary (accessed on 11 June 2023). This is a widely used bioinformatics data set for research on graphical models. The data set offers continuous measurements of expression levels of multiple phosphorylated proteins and phospholipid components in human immune system cells. There are 14 sub-data sets with respect to 14 different biochemical experiments, and the number of data points in each sub-data set ranges from 723 to 917. There are 11 signaling nodes in each sub-data set, and each signaling node represents a phosphorylated protein molecule in the research of the human primary T cell signaling pathway. Over the past two decades, classical biochemistry and genetic analysis have constructed a protein network that can be taken as a ground-truth network. The ground-truth network contains 17 high-confidence causal edges.

### 4.2. Evaluation Metrics

We compared the learned CBN to ground-truth CBN on the for most common graph metrics: (1) Precision; (2) Recall; (3) F1-measure(F1); (4) Structural Hamming Distance (SHD); (5) Time. Specifically, Precision, Recall, and F1 can be defined asfollows:
(11)$$Precision=\frac{TP}{TP+FP},$$
(12)$$Recall=\frac{TP}{TP+FN},$$
(13)$$F1=\frac{2\times Precision\times Recall}{Precision+Recall}.$$

In this paper, we use edges to denote the directed connectivity relationship between two nodes in a CBN. TP denotes the number of edges that exist in both the learned CBN and the ground-truth CBN; FP denotes the number of edges that exist only in the learned CBN compared to the ground-truth CBN; and FN denotes the number of edges that exist in the ground-truth CBN but are not learned. Thus, Precision and Recall range from 0 to 1, and F1 is their harmonic. SHD is the total number of edge additions, deletions, and reversals needed to convert the learned CBN into the ground-truth network. The SHD can be calculated as
(14)SHD=Redu+Miss+Reve,
where Redu represents the number of redundant edges that need to be removed, Miss represents the number of missing edges that need to be added, and Reve represents the number of edges in the opposite direction that need to be reversed. To indicate the time spent by the algorithm in seconds, we use the Time metric.

### 4.3. Contrast Algorithm Introduction and Experimental Setup

To intuitively illustrate the competitiveness of our PACO, we compare with 5 other state-of-the-art or classic algorithms. These algorithms include: continuous optimization for structure learning (NoTears) [23], deep reinforcement learning (DRL) [19], greedy equivalence search (GES) [40], DAG Structure Learning with Graph Neural Networks (DAG-GNN) [18], Artificial Immune Algorithm (AIA) [25]. In addition, to demonstrate the superiority of the parallel strategy of the PACO algorithm, we also compare it with the non-parallel ant optimization colony (ACO) [24] as our ablation experiment. Among the above algorithms, NoTears and DAG-GNN are two that have been successfully applied to learn causal protein signaling networks and achieved good performance on Single-cell data sets. DRL, GES, AIA and ACO are four algorithms to learn causal brain networks that achieved good performance on Smith’s fMRI data set. We use the gCastle toolbox proposed by [41] for the implementation of all publicly available comparison algorithms, and the code is availableat https://github.com/huawei-noah/trustworthyAI (accessed on 11 June 2023). The code for PACO is availableat https://github.com/ZJH66/PACO (accessed on 11 June 2023).

To compare with other algorithms in a fair and appropriate way, we set the parameters of all comparison algorithms to the default values in the citation. The parameters of PACO include the weights for the pheromone trail (α) and for the heuristic information (β), the controls of the pheromone evaporation (ρ), the relative importance of the exploitation versus exploration ($q_{0}$), the number of iterations (NC), and the number of ants (Num). After a large number of experimental tests, we find that the set of parameters α=1.8,β=2,$\;\rho =0.35,\;q_{0}=0.75$ performed well on most of simulation data sets, and the parameters NC and Num are mainly associated with the stability and convergence speed of the algorithm. Thus we test on the simulation data to determine a better parameter configuration of NC and Num for PACO. We find that as Num and NC increase, the learning performance of PACO gets better and better and takes more and more time, and the algorithm achieves the highest accuracy and stabilizes when Num = 20 and NC = 10, from which we determine the final parameters. During all experiments, we employ the control variate technique that the value of a single parameter is changed, while keeping the values of other parameters fixed. The parameter settings of all algorithms are shown in Table 2.

We choose 5 evaluation metrics Precision, Recall, F1,SHD and Time to evaluate the performance of different algorithm and compare PACO with 6 other algorithms using 10 simulation data sets, a set of real fMRI signal data set and a set of Single-cell data sets. To reduce the effect of algorithm randomness on the experimental results, we run all algorithms 10 times on all data sets and take the average. After the validation of the performance and effectiveness of the PACO algorithm, we further compare it on real fMRI signal data sets and real Single-cell data set. The experimental platform is a PC with Intel Core i5-8300, 16 GB RAM, 2.30 GHz CPU, and Windows 10.

### 4.4. The Results of Learning CBNs from Simulation Data Sets

We comprehensively test and compare the above 7 algorithms on the generated 10 simulation data sets, each consisting of a different number of nodes *v* and a different number of sub-data sets *N*. An algorithm performs well when it gets higher values of Precision, Recall and F1 and lower values of SHD and Time. Note that when we test PACO, we use the data set of all the sub-data sets concatenated, the number of colonies *N* in the PACO algorithm is equal to the number of sub-data sets *N* in a simulation data set, and the input of each colony is a sub-data set in a simulation data set. When we test other algorithms, we need to concatenate all the sub-data sets of a simulation data set as the input.

From Table 3, we can find that PACO outperforms the other 6 algorithms in all metrics on the 10 simulation data sets. Specifically, following the two chains Sim1-Sim3-Sim5-Sim7-Sim9 and Sim2-Sim4-Sim6-Sim8-Sim10, each chain has 5 data sets with gradually increasing number of nodes, 5, 10, 30, 50, and 100, respectively, and the difference between the two chains is that each data set in the first chain contains 20 sub-data sets, while each data set in the second chain consists of 50 sub-data sets. Overall, the F1 of all algorithms, including PACO, decreases as the number of nodes *v* and the number of sub-data sets *N* increase. While the Time increases with the number of nodes as well as the number of sub-data sets. When the number of nodes increases to 100, PACO still achieves F1 values of 0.71 and 0.70 on both Sim9 and Sim10 with Time of 5.87 s and 7.66 s, respectively, which is far ahead of the other algorithms and keeps the highest performance. Then we divide the number of nodes into 5 groups to see the impact of the number of sub-data sets on the performance of the algorithm. It is obvious that as the number of sub-data set increases from 20 to 50, the F1 and SHD of most of the compared algorithms get worse and the time cost of all algorithms increases, but the F1 and SHD of PACO get better instead, and we think that the pheromone fusion and CBNs fusion rules we set play a role. In addition, we found that the larger the number of nodes and the larger the number of sub-data set, the more obvious the time advantage of the PACO algorithm. For example, for the Sim10 data set, the number of nodes are 50 and sub-data sets are 100, the time cost of ACO is 13.67 s, while the time cost of PACO is 7.66 s, which is a significant improvement in time performance, which demonstrates the superiority of our parallel strategy on large-sample multi-data set data. For a more visual representation of the stability of the algorithm, we plot the average results on the above 10 simulation data sets in a box plot, as shown in Figure 2.

The above experimental results can fully demonstrate that our proposed algorithm has a more stable performance in all evaluation metrics and a significant improvement in time performance and learning accuracy compared to other algorithms. Next, we will further discuss the performance of the algorithm on real fMRI signal data sets and Single-cell data sets.

### 4.5. The Results of Learning Causal Brain Networks from fMRI Signal Data Sets

In this section, we test the performance of algorithms to learn brain effective connectivity networks from fMRI signal data. The results are shown in Table 4, and the causal brain networks learned by each algorithm are shown in Figure 3. Note that for PACO, we input 50 subjects simultaneously into 50 ant colonies for parallel search, and for the other algorithms, we input 50 sub-data sets in series into the algorithm consecutively.

Combining Table 4 and Figure 3, we can find that DAG-GNN identifies 5 true effective connectivity edges, and the time cost 22.6 s, and the performance of each metric is much lower than other algorithms, which we speculate that this is due to the complex deep learning model used by the algorithm and the assumption that the acyclic graph constraint will lead to a significant performance degradation when generating cyclic graphs. NoTears and DRL identify 7 and 8 true effective connectivity edges, respectively, but the 2 algorithms themselves take more time due to the deep learning and reinforce learning model used. GES identifies 8 true effective connectivity edges, but the Precision is low due to the large number of redundant edges generated; AIA identifies the 7 true effective connectivity edges, and performance is moderate in all metrics, with no outstanding advantages. ACO identifies 9 true effective connectivity edges, Recall, Precision and F1 are 0.82. PACO identifies 9 true effective connectivity edges and outperforms ACO on all evaluation metrics. Moreover, ACO consumes 0.98 s while PACO consumes 0.49 s, which is almost a times improvement in time performance, which proves that our pheromone fusion and CBNs fusion mechanism and parallel search strategy have achieved significant results. In summary, PACO can learn causal brain networks more accurately and efficiently.

### 4.6. The Results of Learning Causal Protein Signaling Networks from Simulation Data Sets

We further validate the performance of the PACO algorithm on the Single-cell data set. For PACO, we input 14 sub-data sets simultaneously into 14 ant colonies for parallel search, and for other algorithms, we input 14 sub-data sets in series into the algorithm consecutively. The results are shown in Table 5, and the causal protein signaling networks learned by each algorithm are shown in Figure 4.

Combining Table 5 and Figure 4, we can find that PACO outperforms all other algorithms in all evaluation metrics, where SHD is 15 with both NoTears and DRL, but PACO is far ahead of NoTears and DRL in terms of time. NoTears, DRL, and DAG-GNN are at the same level of accuracy in learning causal protein signaling networks, and all learn nearly half of the connected edges of protein signals correctly, but all three algorithms have a significant disadvantage in time performance due to the relatively complex neural networks and deep learning models they all use. The Recall of GES is 0.41, but the Precision is 0.2, which leads to an F1 of only 0.27 and a SHD of up to 30, obviously learning a large number of redundant signal connection edges. AIA and ACO algorithms also have the same characteristics as GES algorithm in terms of evaluation metrics. The above three algorithms have short running time but less accuracy. PACO has an obvious advantage in accuracy with Precision, Recall and F1 of 0.53, which proves that our pheromone fusion and CBNs fusion strategy can improve the accuracy of the algorithm in learning CBNs, and also has an obvious advantage in time, which proves that our parallel strategy is effective. In summary, PACO can learn causal protein signaling networks accurately and efficiently.

## 5. Conclusions

In this paper, we introduce a novel parallel ant colony optimization algorithm (PACO) for learning CBNs from biological signal data. Our experiments on the generated simulation data set, the real fMRI signal data set and the real Single-cell data set show that PACO has significant improvements in accuracy performance and time performance compared to the state-of-the-art algorithm and the advantage of this method will be more obvious when dealing with large scale multiple data sets. Compared with the non-parallel ant colony optimization (ACO) algorithm, PACO shows a significant improvement in accuracy and time performance, which proves the effectiveness of the parallel strategy and pheromone fusion and CBNs fusion mechanism. In future work, we plan to further investigate the acyclicity constraint during parallel ant colony optimization to reduce the search space to improve the performance.

## Figures and Tables

**Figure 1 bioengineering-10-00909-f001:**
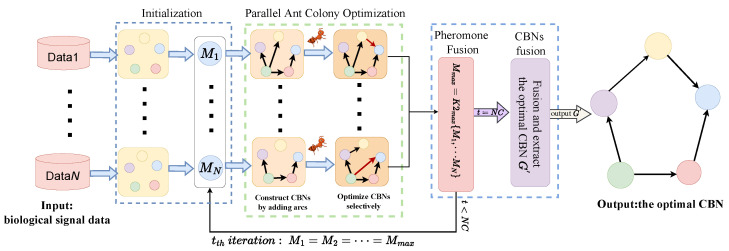
The flowchart of the PACO algorithm.

**Figure 2 bioengineering-10-00909-f002:**
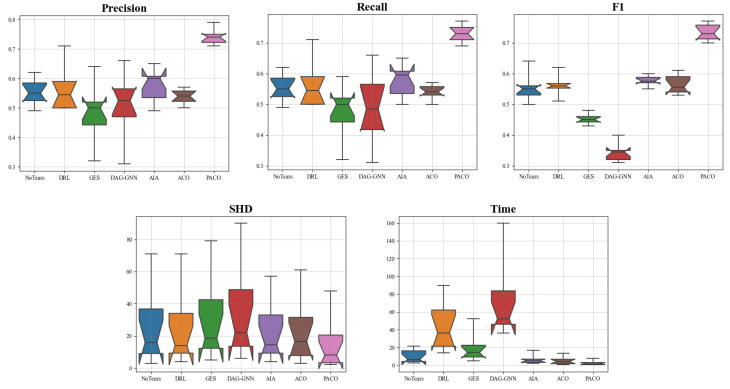
The average results of 7 algorithms on 10 simulation data sets corresponding to 5 metrics. The box represents the middle 50% of the data, with a horizontal line inside the box representing the mean. The horizontal axis represents 7 algorithms, and the vertical axis is the value of the evaluation metrics.

**Figure 3 bioengineering-10-00909-f003:**
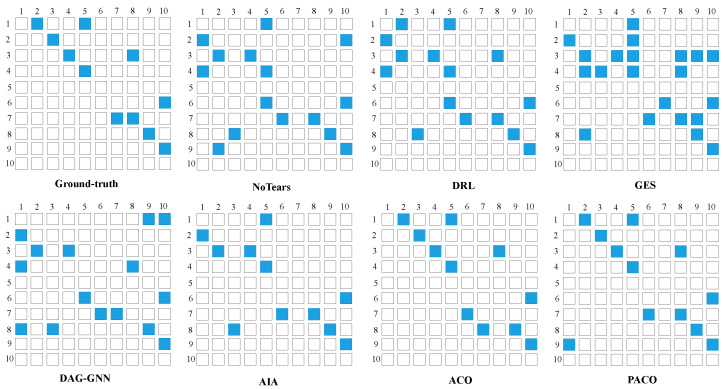
The ground-truth and the causal brain networks learned from fMRI signal data set. The horizontal and vertical coordinates indicate the corresponding brain regions of interest and the blue grid indicates effective connectivity between the two corresponding brain regions.

**Figure 4 bioengineering-10-00909-f004:**
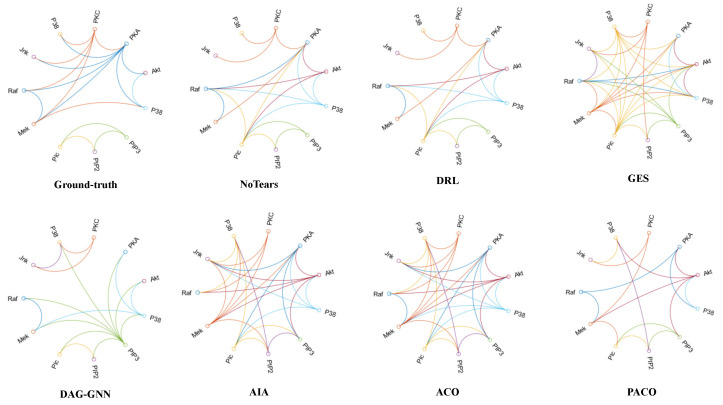
The ground-truth and the causal protein signaling networks learned from Single-cell data. The connection curves in the graph represent the signaling networks between the 11 phosphorylated protein biomolecules.

**Table 1 bioengineering-10-00909-t001:** The introduction of different CBN learning methods.

Category	Methods	Years	Category	Methods	Years
Causal Brain Network	spectral Dynamic Causal Modeling (spDCM)	2014 [22]	Causal Protein Signaling Network	Continuous Optimization (NoTears)	2018 [23]
	Ant Colony Optimization (ACO)	2016 [24]		Graph Neural Network (DAG-GNN)	2019 [18]
	Artificial Immune Algorithm (AIA)	2016 [25]		Reinforcement Learning (RL)	2019 [26]
	Generative Adversarial Network (GAN)	2020 [20]		Three Track Neural Network (TTNN)	2021 [27]
	Large-scale Dynamic Causal Mode (PEB)	2020 [28]		Latent Factor Causal Models (LFCMs)	2022 [29]
	Recurrent Generative Adversarial Network (RGAN)	2021 [21]		Truncated Matrix Power Iteration (TMPI)	2022 [16]
	Deep Reinforcement Learning (DRL)	2022 [19]		BN with Pruning Strategies (CO-CDG)	2022 [15]
	Amortization Transformer (AT-EC)	2023 [30]		Deconfounded Functional Structure Estimation (DeFuSE)	2023 [31]

**Table 2 bioengineering-10-00909-t002:** Parameter settings of 7 algorithms.

Algorithms	Parameters
NoTears [23]	$\lambda_{1}=0.1,\;max_{iter}=100,\;threshold$ = 0.3
DRL [19]	epoch=50, α=0.99
GES [40]	k=0.01, N=10
DAG-GNN [18]	epoch=300, η = 10, γ=1
AIA [25]	$P_{s}=0.5,\;P_{c}=0.6,\;P_{m}=0.4,$
	T=150, N=80, M=70
ACO [24]	α=1.8, β=1.5, ρ=0.6,
	$q_{0}=1.2,\;NC=10,\;Num=30$
PACO	$\alpha =1.2,\;\beta =2,\;\rho =0.35,\;q_{0}=0.75$
	NC=10, Num=20

**Table 3 bioengineering-10-00909-t003:** Comparisons of 7 algorithms on 10 simulation data sets.

Data (*v*,*N*)	Metrics	Algorithms						
		NoTears	DRL	GES	DAG-GNN	AIA	ACO	PACO
Sim1(5,20)	Precision	0.62	0.61	0.55	0.38	0.60	0.57	**0.79**
	Recall	0.66	0.51	0.35	0.27	0.60	0.62	**0.75**
	F1	0.64	0.56	0.45	0.32	0.60	0.59	**0.77**
	SHD	3	4	5	6	4	3	**2**
	Time (s)	2.76	14.12	5.19	36.12	1.98	0.79	**0.51**
Sim2 (5,50)	Precision	0.59	0.62	0.51	0.35	0.60	0.56	**0.75**
	Recall	0.61	0.52	0.35	0.35	0.50	0.62	**0.77**
	F1	0.60	0.57	0.43	0.35	0.55	0.59	**0.76**
	SHD	3	4	6	7	5	3	**2**
	Time (s)	4.51	19.25	10.32	45.34	2.73	1.42	**0.78**
Sim3 (10,20)	Precision	0.59	0.62	0.53	0.40	0.60	0.56	**0.77**
	Recall	0.53	0.51	0.43	0.30	0.54	0.56	**0.75**
	F1	0.56	0.56	0.48	0.35	0.57	0.56	**0.76**
	SHD	9	9	12	13	10	7	**4**
	Time (s)	4.12	17.22	8.91	39.73	2.36	1.34	**0.68**
Sim4 (10,50)	Precision	0.57	0.55	0.49	0.38	0.60	0.52	**0.75**
	Recall	0.55	0.51	0.45	0.40	0.52	0.54	**0.75**
	F1	0.56	0.53	0.47	0.39	0.56	0.53	**0.75**
	SHD	9	10	13	15	9	10	**3**
	Time (s)	7.89	25.64	13.67	48.26	4.41	2.86	**0.91**
Sim5 (30,20)	Precision	0.55	0.62	0.54	0.37	0.58	0.53	**0.73**
	Recall	0.50	0.52	0.34	0.27	0.59	0.53	**0.71**
	F1	0.53	0.57	0.44	0.32	0.59	0.53	**0.72**
	SHD	17	15	19	23	14	17	**9**
	Time (s)	7.54	24.33	14.98	54.66	4.61	3.64	**1.56**
Sim6 (30,50)	Precision	0.56	0.63	0.55	0.39	0.61	0.54	**0.75**
	Recall	0.56	0.61	0.35	0.31	0.54	0.56	**0.73**
	F1	0.56	0.62	0.45	0.35	0.58	0.55	**0.74**
	SHD	15	13	18	21	15	16	**7**
	Time (s)	9.14	30.55	7.16	50.37	5.69	4.79	**1.95**
Sim7 (50,20)	Precision	0.55	0.61	0.48	0.36	0.62	0.60	**0.73**
	Recall	0.53	0.51	0.44	0.27	0.52	0.62	**0.71**
	F1	0.54	0.56	0.46	0.31	0.57	0.61	**0.72**
	SHD	36	34	43	48	33	30	**19**
	Time (s)	13.36	45.62	18.96	75.33	5.97	5.65	**2.73**
Sim8 (50,50)	Precision	0.54	0.59	0.51	0.38	0.63	0.57	**0.72**
	Recall	0.52	0.53	0.32	0.27	0.54	0.61	**0.70**
	F1	0.53	0.56	0.43	0.32	0.58	0.59	**0.71**
	SHD	37	34	41	49	33	32	**21**
	Time (s)	16.64	57.89	23.67	86.51	7.15	7.11	**3.15**
Sim9 (100,20)	Precision	0.52	0.56	0.55	0.38	0.61	0.55	**0.71**
	Recall	0.52	0.54	0.35	0.42	0.52	0.53	**0.71**
	F1	0.52	0.55	0.45	0.40	0.57	0.54	**0.71**
	SHD	68	60	79	86	57	61	**48**
	Time (s)	32.76	98.75	41.63	139.87	10.36	10.88	**5.87**
Sim10 (100,50)	Precision	0.49	0.51	0.45	0.35	0.58	0.54	**0.71**
	Recall	0.51	0.51	0.47	0.33	0.60	0.54	**0.69**
	F1	0.50	0.51	0.46	0.34	0.59	0.54	**0.70**
	SHD	71	71	77	90	55	60	**46**
	Time (s)	42.36	135.5	52.36	159.62	16.98	13.67	**7.66**

The bold values indicate that the algorithm achieved the best results.

**Table 4 bioengineering-10-00909-t004:** Comparisons of 7 algorithms on the fMRI signal data set.

Algorithms	Precision	Recall	F1	*SHD*	*Time* (s)
NoTears	0.47	0.64	0.54	9	9.20
DRL	0.57	0.73	0.64	6	16.10
GES	0.38	0.73	0.50	13	2.91
DAG-GNN	0.33	0.45	0.38	13	22.60
AIA	0.64	0.64	0.64	5	1.35
ACO	0.82	0.82	0.82	3	0.98
PACO	**0.83**	**0.91**	**0.87**	**2**	**0.49**

The bold values indicate that the algorithm achieved the best results.

**Table 5 bioengineering-10-00909-t005:** Comparisons of 7 algorithms on the Single-cell data set.

Algorithms	Precision	Recall	F1	*SHD*	*Time* (s)
NoTears	0.44	0.47	0.45	**15**	35.30
DRL	0.47	0.47	0.47	**15**	63.20
GES	0.20	0.41	0.27	30	11.60
DAG-GNN	0.44	0.41	0.42	17	122.70
AIA	0.19	0.35	0.24	31	3.80
ACO	0.25	0.47	0.33	26	3.10
PACO	**0.53**	**0.53**	**0.53**	**15**	**1.90**

The bold values indicate that the algorithm achieved the best results.

## Data Availability

Not applicable.

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
