# Peer review of "Learning Causal Biological Networks with Parallel Ant Colony Optimization Algorithm"

_bioengineering, 2023, doi:10.3390/bioengineering10080909_

Round 1

Reviewer 1 Report

To enhance performance and decrease estimation time, the Parallel Ant Colony Optimization Algorithm was used in this paper to learn causal biological networks. It is a well-written paper and there is potential for improvement:

1- TP, FP, TN, and FN need to be redefined for the context of CBN for readers.

2- It is not clear if the 6 comparison methods such as NoTears developed by authors or another source was used for CBN estimation. please clarify.

3- I am concerned about the reproducibility of results by other researchers and invite authors to provide source code (e.g., through Github) for their proposed method and other comparison methods (if developed by them).

There is room for improvement in Quality of English Language. 

Reviewer 2 Report

1.The authors proposed a novel algorithm (the parallel ant colony optimization - PACO) for Learning Causal Biological Networks. The algorithm consists of three phases: (i) initialization, (ii) parallel ant colony optimization, and (iii) information fusion.

2.In addition a new information fusion mechanism is suggested for merging and updating the pheromones of all colonies after the completion of the same iteration of all ant colonies. Further the fusion results are used as the initial pheromones of the colonies in the next iteration.

3.The description of the author approach is presented at a very good level. The flowchart of the suggested PACO algorithm is interesting and very useful.

4.The described extensive experimental results are based on simulation data sets (two real-world data sets): (a) the fMRI signal data set and (b) the Single-cell data set. The experimental results show that suggested PACO has a good level of accuracy and time performance (i.e., algorithmic complexity).

5.Generally, the paper material is interesting and useful from the theoretical viewpoint and from  the practical view point. The paper material is presented at a good level (all parts). The paper can be accepted (as is).

6.In sections ‘introduction’ and ‘related works’ it may be reasonable and useful for readers to add a structural description of the related works on the basis a table and/or scheme (e.g., framework as taxonomy).

7.From  the viewpoint of information fusion approaches it may be interesting for authors to look for various fusion methods, for example, in journal ‘Information Fusion’ (in the future).

Reviewer 3 Report

1. Implementation details for the conducted algorithm is missing as well as the simulation setup

2. It is not clear how the proposed algorithm terminates

3. The authors should discuss how they selected the parameters of table 1

Round 2

Reviewer 3 Report

The paper can be accepted in the current form